# Research on the High Sensitivity Detection Method of Carbon Nanotube/Polydimethylsiloxane Composites Structure

**DOI:** 10.3390/mi13050719

**Published:** 2022-04-30

**Authors:** Lishuang Liu, Ruirong Wang, Hao Guo, Jinping Liu, Xin Li, Yue Qin, Jun Tang

**Affiliations:** 1Key Laboratory of Instrumentation Science and Dynamic Measurement, School of Instrument and Electronics, North University of China, Taiyuan 030051, China; wangruirong@yeah.net (R.W.); guohaonuc@163.com (H.G.); s1906033@163.com (J.L.); lixinxjx@163.com (X.L.); qytgyx@163.com (Y.Q.); tangjun@nuc.edu.cn (J.T.); 2Department of Electronic Engineering, Taiyuan Institute of Technology, Taiyuan 030008, China

**Keywords:** carbon nanotube/polydimethylsiloxane (CNT/PDMS), force-sensitive test, high sensitivity

## Abstract

In this paper, a carbon nanotube (CNT)/polydimethylsiloxane (PDMS) composite force-sensitive structure with good flexibility is proposed and fabricated, and the measurement of *scanning electron microscopy* (SEM) and Raman are carried out. The equivalent circuit of force-sensitive test of structure is performed and analyzed under direct current (DC) and alternating current (AC) conditions. Under AC conditions, experimental results further show that the sensitivity and sensitivity factors of force-sensitive structures are 0.15 KPa^−1^ and 2.17 in the pressure range of 600–1000 KPa compressive stress and 20–50% tensile stress, respectively. These results are increased by 36.4% and 38.2% compared to the results of compressive stress (0.11 KPa^−1^) and tensile stress (1.57) under DC conditions, respectively. It shows that the carbon nanotube/PDMS composite has higher test accuracy under AC conditions.

## 1. Introduction

Flexible pressure sensors can adhere to irregular surfaces perfectly due to their soft characteristics. Thus, they can be used to monitor external mechanical signals and are widely used in the fields of wearable electronics [1,2,3], robots [4], medical monitoring [5,6], human–machine interaction [7,8], and so on.

According to the different signal transmission modes, flexible pressure sensors can be divided into three types: piezoresistive [9,10], capacitive [11,12], and piezoelectric [13]. Among them, piezoresistive sensors are widely studied because of their simple preparation, low cost, convenient signal acquisition, high-pressure measurement range, and high sensitivity [14,15,16].

Currently, the sensitivity test of flexible pressure sensors is mainly carried out through the change of resistance under stress [17,18]. H.B. Yao [19] et al. designed a grapheme-polyurethane sponge sensor with a high-pressure sensitivity of 0.03 KPa^−1^ in the pressure range of 2–10 KPa by testing the change of resistance under different pressures. A.D. Smith [20] et al. proposed a suspended graphene film piezoresistive sensor with sensitivity of 2.25 × 10^−3^ KPa^−1^ in the pressure range of 0–100 KPa. S. Chun [21] et al. designed a double-layer graphene high-sensitivity pressure sensor structure with estimated pressure sensitivity of 0.034 KPa^−1^ for the linear response range of 1–8 KPa. Thus, the test accuracy is not high by means of the change of resistance to test the sensitivity of force-sensitive structure, which further affects the improvement of sensitivity. C. Sungwoo [11] proposed CNT-sheet-film-based pressure sensors with a sensitivity factor of 0.02–0.04% in the pressure range of 20–40 KPa.

In this paper, a CNT/PDMS composites force-sensitive structure with good flexibility is fabricated. In order to test the sensitivity, the force-sensitive test equivalent circuit of the structure is theoretically analyzed under DC and AC conditions. It is obtained that the impedance change of the force-sensitive structure under AC conditions is more accurate than that under DC conditions. Further, the force-sensitive characteristics of CNT/PDMS nanocomposite force-sensitive structures were tested under different DC and AC conditions. The results show that the sensitivity and sensitivity factor of the force-sensitive structure under the action of 600–1000 KPa compressive stress and 20–50% tensile stress are 0.15 KPa^−1^ and 2.17 under the AC condition, respectively, which are 36.4% and 38.2% higher than 0.11 KPa^−1^ and 1.57 under the DC condition. It shows that the carbon nanotube (CNT)/PDMS composites have higher test accuracy under AC conditions.

## 2. Theoretical Analysis

Under the DC condition, the stress exerted on the nanocomposite structure will cause the change of position and shape of CNT in the polymer, and the distance between CNT, which results in the change of conductive network and resistance. The deformation will lead to a change of resistance. The equivalent circuit is shown in Figure 1a,b. The change of resistance is defined as:(1)$$\Delta R=R-R_{0}$$
where *R*_0_ is the resistance value of the structure without stress and *R* is the resistance value of the structure under stress.

Under AC conditions, there will be a resultant change in resistance as well as a change of capacitance between the conductive grids when stress is applied to the CNT/PDMS force-sensitive structure. Thus, the total changes are replaced by impedance equivalent. The equivalent circuit is shown in Figure 1c,d. The change of impedance can be expressed as:(2)$$\Delta Z=Z-Z_{0}=(R-jX_{C})-(R_{0}-jX_{C0})\;=(R-R_{0})-(jX_{C}-jX_{C0})\;=\Delta R-j\Delta X_{C}$$
where *Z*_0_ is the impedance value of the structure without applied stress and *Z* is the impedance value of the structure under stress.

By comparing Equations (1) and (2), it shows that the change of impedance can be divided into two parts: the change of resistance and capacitive reactance after stress is applied under AC conditions, which is more accurate than the change of single resistance under DC conditions. Therefore, the measurement accuracy of the sensitivity of force-sensitive structures will be improved by measuring the change of impedance.

## 3. Structure Fabrication

### 3.1. Materials

Carbon nanotube (CNT) was purchased from Shenzhen suiheng Technology Co., Ltd. Shenzhen, China. The detailed specifications of CNTs are a length of 2–8 μm, an inner diameter of 5–8 nm, an outer diameter of 10–15 nm, and a purity of 98%. Polydimethylsiloxane (PDMS) and curing agent were purchased from Dow Corning, Midland, MI, USA.

### 3.2. Processing of Sample

Firstly, 0.54 g CNT were weighed as conductive filler, and a small amount of alcohol was added to them and then vibrated for one hour in an ultrasonic cell crusher to make carbon nanotubes evenly dispersed in alcohol. Then, 10 ml PDMS were added to it for mechanical stirring. After mixing, the curing agent was added and stirred to form a homogenous slurry. Finally, the slurry was cast into a mold. Then, the mold was put into the vacuum oven for about 1h to remove the bubbles in the sample. Subsequently, the mold was cured on the heating table at a temperature of 80 °C for 2 h. To obtain the CNT/PDMS force-sensitive structure, the as-prepared mold was cut into the size of 2 cm × 1 cm for performance measurement. The processing flow of the sample is shown in Figure 2. Figure 3a show that the processed sample has good flexibility. The conductivity of the CNT/PDMS composites structure was 38 S/m. The cross-section of the composite was observed by scanning electron microscope, as shown in Figure 3b. In Figure 3b, carbon nanotubes are evenly distributed in the CNT/PDMS composites. Figure 3c show the Raman spectrum of CNT/PDMS.

## 4. Force-Sensitive Characteristic Test

The force-sensitive characteristics of the prepared CNT/PDMS force-sensitive structure under DC conditions were tested. The test principle is shown in Figure 4a. Figure 5a show the change results of the relative resistance of CNT/PDMS force-sensitive structure with pressure in the range of 0–1000 KPa. The results show that the resistance change rate increases as the function of pressure. In the pressure range of 600–1000 KPa, the sensitivity of the CNT/PDMS force-sensitive structure was 0.11 KPa^−1^ (the sensitivity calculation formula is (*R* − *R*_0_)/*R*_0_/Δ*P*, *R* is the resistance of the structure after pressure is applied, *R*_0_ is the resistance value of the structure without pressure, and Δ*P* is the change of corresponding pressure). Figure 5b show the change of relative resistance of the CNT/PDMS force-sensitive structure within 50% tensile strain, and it is indicated that the sensitivity of the CNT/PDMS force-sensitive structure is 1.57 within 20–50% tensile strain (the sensitivity calculation formula is (*R* − *R*_0_)/R_0_/strain, *R* is the resistance of the structure under tensile strain, and *R*_0_ is the resistance of the structure without tensile strain).

The force-sensitive characteristics of the prepared CNT/PDMS were tested under AC conditions. The testing principle is shown in Figure 4b, and the test results are shown in Figure 6. Figure 6a show the change in the relative impedance of the CNT/PDMS force-sensitive structure with pressure in the range of 0–1000 KPa. The change rate of impedance increases as the increase of pressure. The sensitivity of CNT/PDMS was 0.15 KPa^−1^ in the pressure range of 600–1000 KPa (the sensitivity is calculated by (*Z* − *Z*_0_)/*Z*_0_/Δ*P*, where *Z* is the impedance of the structure after pressure application, *Z*_0_ is the impedance value of the structure without pressure application, and Δ*P* is the change of corresponding pressure). Figure 6b show the change of relative impedance of CNT/PDMS force-sensitive structures within the range of 50% tensile strain. It can be seen that the sensitivity of CNT/PDMS nanocomposites was 2.17 within the range of 20–50% tensile strain from Figure 6b (the sensitivity calculation formula is (*Z* − *Z*_0_)/*Z*_0_/strain, *Z* is the impedance value of the structure under tensile strain, and *Z*_0_ is the impedance value of the structure without tensile strain). 

From Figure 5 and Figure 6, the sensitivity and sensitivity factors of the CNT/PDMS force-sensitive structure under AC conditions are 0.15 KPa^−1^ and 2.17, respectively, and 0.11 KPa^−1^ and 1.57, respectively, under DC conditions in the pressure range of 600–1000 KPa and 20–50% tensile strain. The comparison results show that the pressure sensitivity and tensile strain sensitivity factors under AC conditions were increased by 36.4% and 38.2%, respectively. Figure 7 show the comparison of relative impedance under AC and resistance change of the CNT/PDMS composite under DC when pressure and tensile strain were applied, which shows that the pressure sensitivity and tensile strain sensitivity factors under AC conditions are higher than in DC. Table 1 show the sensitivity comparison between our flexible sensors test results and other sensor test results. From the comparison results, it can be seen that the sensitivity of the test method in this paper is improved by at least three times.

In order to further test the force-sensitive characteristics of the CNT/PDMS force-sensitive structure under AC conditions, the force-sensitive characteristics of the composites structure under different tensile strains at fixed frequency are tested, and the results are shown in Figure 8. Figure 8a is the schematic test diagram; Figure 8b,c show the frequency of the force-sensitive structure decreases under different tensile strains, and the frequency is 14.7 kHz when without stretched, the frequency decreases to 11.5 kHz after 50% tensile strain and the corresponding frequency decreases by about 22.5%. This could be explained by *f* = 1/2*πRC*, the resistance increases, the capacitance decreases under tensile strain, and the change of resistance is greater than that of capacitance, so the frequency decreases. Under 50% tensile strain, the frequency is reduced by 3.3 kHz, indicating that the strain detection of force-sensitive structure can reach 170 με, which is calculated according to the formula $\frac{1}{\left(\Delta f/\Delta \varepsilon \right)}$, Δf is the variation of frequency under different tensile strain, Δε is the variation of tensile strain. Under the action of tensile strain, the change rate of resistance and capacitance are increasing simultaneously and further result in the increase of the change rate. Therefore, the frequency shows a significant change with the increase of tensile strain. According to Figure 5b, it is known that the strain detection of force-sensitive structure is 8000 με under DC, which is calculated according to the formula $\frac{1}{\left(\Delta R/\Delta \varepsilon \right)}$. The results show that the frequency measurement under AC conditions has higher sensitivity. Moreover, the resistance measurement can not be compatible with the AC system, but the frequency measurement method can be compatible with the AC system, such as electromagnetic field, which has a wider range of applications and prospects.

## 5. Conclusions

In this paper, a CNT/PDMS force-sensitive structure was designed and fabricated. The structural morphology test and electrical performance tests show that the machined force-sensitive structure has good conductivity and flexibility. The equivalent circuit of stress-sensitivity test of the CNT/PDMS nanostructures was theoretically analyzed under DC and AC conditions. Further experimental results show that under AC conditions, the sensitivity and sensitivity factors of stress-sensitive structures in the pressure range of 600–1000 KPa and 20–50% tensile strain are 0.15 KPa^−1^ and 2.17, respectively, which are 36.4% and 38.2% higher than that under DC conditions. It shows that the CNT/PDMS composite has higher test accuracy under AC conditions.

## Figures and Tables

**Figure 1 micromachines-13-00719-f001:**
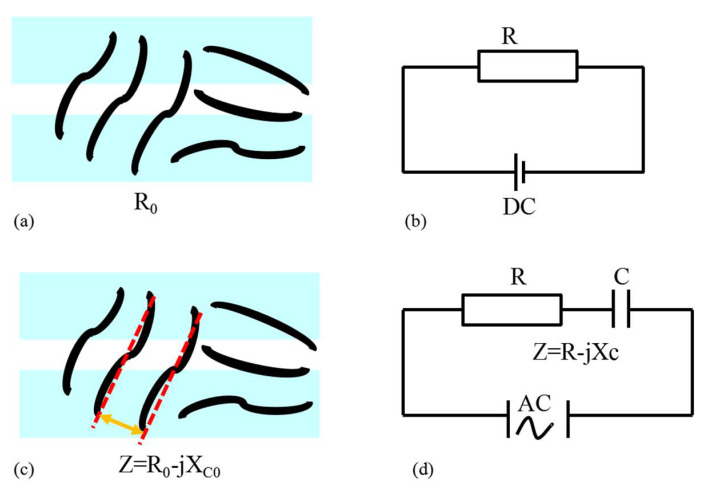
The equivalent circuit of CNT/PDMS force-sensitivity test (**a**,**b**) DC condition (**c**,**d**) AC condition.

**Figure 2 micromachines-13-00719-f002:**
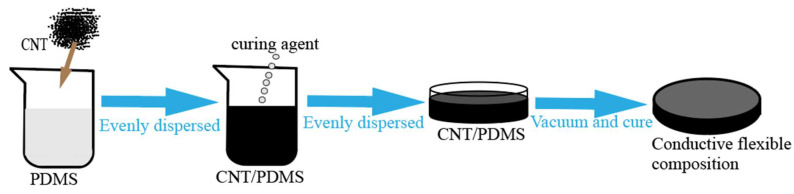
Processing of CNT/PDMS force-sensitive structure.

**Figure 3 micromachines-13-00719-f003:**
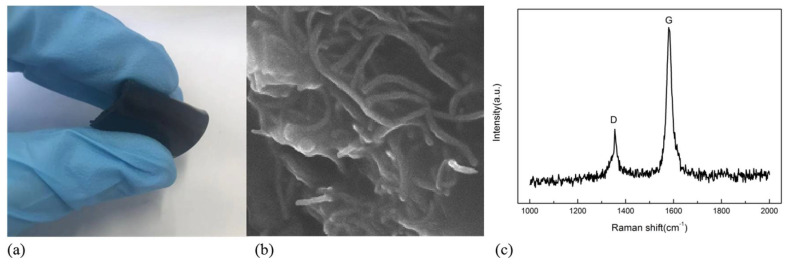
CNT/PDMS (**a**) physical diagram of force-sensitive structure (**b**) SEM diagram (**c**) test results of Raman spectrum.

**Figure 4 micromachines-13-00719-f004:**
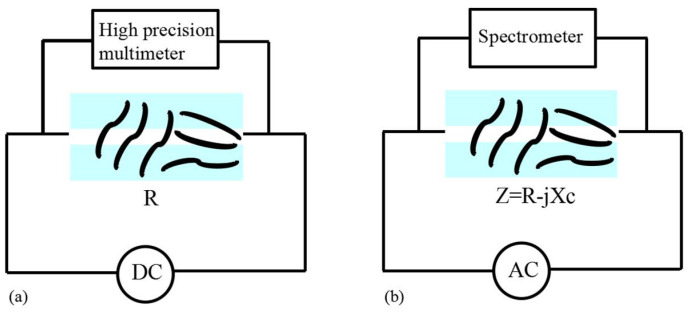
Schematic diagram of sensitivity test of CNT/PDMS force-sensitive structure (**a**) under DC conditions (**b**) under AC conditions.

**Figure 5 micromachines-13-00719-f005:**
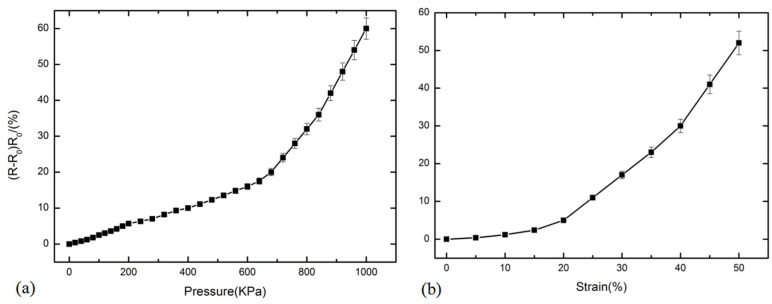
The relative resistance change of CNT/PDMS composites under (**a**) pressure and (**b**) tensile strain.

**Figure 6 micromachines-13-00719-f006:**
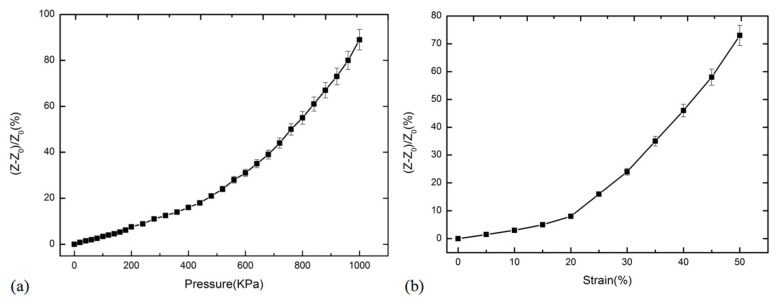
The relative impedance change of CNT/PDMS composite under (**a**) pressure and (**b**) tensile strain.

**Figure 7 micromachines-13-00719-f007:**
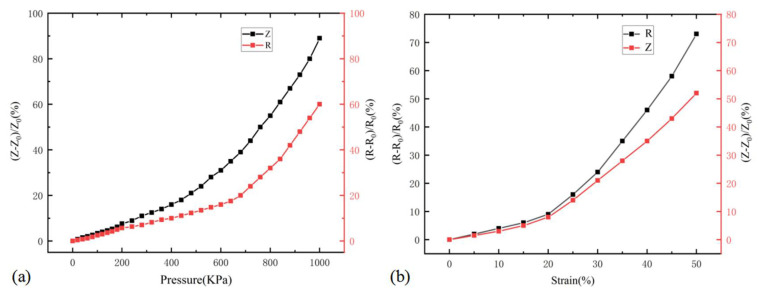
The comparison of relative impedance and resistance change of CNT/PDMS composite under (**a**) pressure and (**b**) tensile strain.

**Figure 8 micromachines-13-00719-f008:**
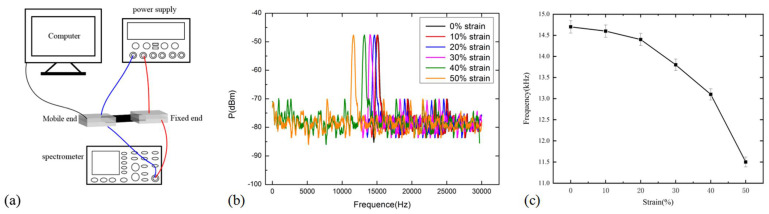
The sensitive characteristic test of force-sensitive structure (**a**) the schematic test diagram (**b**) and (**c**) the result of frequency variation under different tensile strains.

**Table 1 micromachines-13-00719-t001:** The comparison between our flexible sensors test results and other sensor test results.

Source	Key Material	Mechanical Component	Transduction Principles	Sensitivity	Range
H.B.Yao [19]	graphene-polyurethane sponge	Pressure	Piezoresistivity	0.03 KPa^−1^	2–10 KPa
A.D. Smith [20]	graphene membranes	Pressure	Piezoresistivity	2.25 × 10^−^^3^ KPa^−1^	0–100 KPa
S. Chun [21]	double-layer graphene	Pressure	Piezoresistivity	0.034 KPa^−1^	1–8 KPa
C. Sungwoo [11]	CNT-sheet-film	Pressure	Capacitance	0.02–0.04%	20–40 KPa
This work	CNT/PDMS	Pressure Strain	Impedance Impedance	0.11 KPa^−1^ 1.57	600–1000 KPa 20–50%

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
