# Peer review of "Research on the High Sensitivity Detection Method of Carbon Nanotube/Polydimethylsiloxane Composites Structure"

_micromachines, 2022, doi:10.3390/mi13050719_

Round 1
Reviewer 1 Report
I recommend accept it for publication after addressing the following points,
In this work author focused on highly sensitive detection methodology by preparing Carbon nanotube/polydimethylsiloxane composite force sensitive structure. Here, experimental results are sound with good discussions. I recommend its publication with minor revision and re-review as listed below.
Minor comments
- English needs some improvement.
- The title is not sound; the author should revise the title.
- Abstract needs more quantitative information. Also, the author should write the keywords in full form.
- The introduction needs more quantitative information.
- Purity of the used chemicals is missing?
- Recheck all the abbreviations, equations, and figures (there captions).
- The figure resolution is poor; the author should improve it.
Major comments
- The conductivity and resistivity of the prepared material are missing in the manuscript author should do to improve the strength of the results
- What is the active surface area of the fabricated material?
- Error bar missing in graphs
- How relative resistance change calculated
- Before using carbon nanotube did you conducted any pretreatment
13. in introduction flexible sensor uses must be included
14. authors must include more justification about the result
15. Comparison table missing
Author Response
Submission Number: micromachines-1686652
Dear Editor:
We would like to thank you for giving us a chance to modify the paper (micromachines-1686652) entitled " Study on high sensitivity detection method of Carbon nanotube / polydimethylsiloxane force sensitive structure ", and thank the reviewers’ constructive suggestions for us to improve quality of the paper. We have studied all comments carefully and made a revision, which we wish to meet with approval. The changes are marked with the yellow background in the revised manuscript. The following text lists the original comments (black sentences) and the replies (red sentences). Please feel free to contact me if any additional information is needed.
Sincerely yours,
Professor Lishuang Liu
Email: [email protected]
--------------------------------------------------------------------
Reviewers' comments:
Reviewer: 1
Minor comments
1/English needs some improvement.
Reply
Thanks for the reviewer`s suggestion. We have checked and polished the language throughout the manuscript.
2/The title is not sound; the author should revise the title.
Reply
We have modified the title as “Research on the high sensitivity detection method of Carbon nanotube / polydimethylsiloxane composites structure”.
3/Abstract needs more quantitative information. Also, the author should write the keywords in full form.
Reply
We have supplemented the quantitative information in abstract, and write the keywords in full form.
4/The introduction needs more quantitative information.
Reply
We have supplemented the quantitative information in introduction.
5/Purity of the used chemicals is missing?
Reply
The purity of Carbon nanotube (CNT) is 98%.
6/Recheck all the abbreviations, equations, and figures (there captions).
Reply
We have rechecked all the abbreviations, equations, and figures
7/The figure resolution is poor; the author should improve it.
Reply
We have redrawn all the pictures.
8/The conductivity and resistivity of the prepared material are missing in the manuscript author should do to improve the strength of the results
Reply
The conductivity of the prepared material are 38S/m. We have added in the paper.
9/What is the active surface area of the fabricated material?
Reply
The active surface area of the fabricated material 2cm×1cm
10/Error bar missing in graphs
Reply
We have added the error bar in graphs.
11/How relative resistance change calculated
Reply
The sensor tensile strain test platform was used to test the change of resistance under the action of tensile strain. The JT-1500 control test platform of temperature and pressure composite environment was used to test the change of resistance under the action of pressure at room temperature.
12/Before using carbon nanotube did you conducted any pretreatment
Reply
We added the alcohol to carbon nanotube, so the dispersity will be better. After mixing PDMS, heat up to evaporate the alcohol.
13/in introduction flexible sensor uses must be included
Reply
We have added the flexible sensor uses in introduction.
14/ authors must include more justification about the result
Reply
We have added more justification about the result in paper and marked with the yellow background in the revised manuscript.
15/Comparison table missing
Reply
We have added the comparison table in paper.

Reviewer 2 Report
This paper looks novel in terms of study presented but I would like to suggest some minor changes before final acceptance.
the authors need to add some latest literature and add it to introduction. The problems which are being addressed in this paper should be more clear.
how the present methodology provides better result than existing method?
The few grammatical errors could be corrected before final acceptance.
I can see some typo errors too, must be improved.
Author Response
Submission Number: micromachines-1686652
Dear Editor:
We would like to thank you for giving us a chance to modify the paper (micromachines-1686652) entitled " Study on high sensitivity detection method of Carbon nanotube / polydimethylsiloxane force sensitive structure ", and thank the reviewers’ constructive suggestions for us to improve quality of the paper. We have studied all comments carefully and made a revision, which we wish to meet with approval. The changes are marked with the yellow background in the revised manuscript. The following text lists the original comments (black sentences) and the replies (red sentences). Please feel free to contact me if any additional information is needed.
Sincerely yours,
Professor Lishuang Liu
Email: [email protected]
--------------------------------------------------------------------
Reviewers' comments:
Reviewer: 2
1/ the authors need to add some latest literature and add it to introduction. The problems which are being addressed in this paper should be more clear.
Reply
Thanks for the reviewer`s suggestion. We have added some latest literature in introduction.
2/ how the present methodology provides better result than existing method?
Reply
We added Table 1 shows the sensitivity comparison between our flexible sensors test results and other sensor test results, which show that the sensitivity of the test method in this paper is improved by at least three times.
3/The few grammatical errors could be corrected before final acceptance.
Reply
Thanks for the reviewer`s suggestion. We have checked and polished the language throughout the manuscript.
4/I can see some type errors too, must be improved.
Reply
We have corrected the type errors.
